# Self-Organizing Maps Analysis of Chemical–Mineralogical Gold Ore Characterization in Support of Geometallurgy

**Fabrizzio Rodrigues Costa** [1,*]**, Cleyton de Carvalho Carneiro** [1,2] **and Carina Ulsen** [1,3]

1. Department of Mining and Petroleum Engineering, Universidade de São Paulo, Escola Politécnica, São Paulo 05508030, Brazil
2. InTRA—Integrated Technologies for Rock and Fluid Analysis, Universidade de São Paulo, Escola Politécnica, Santos 11013552, Brazil
3. Technological Characterization Laboratory, Department of Mining and Petroleum Engineering, Universidade de São Paulo, Escola Politécnica, São Paulo 05508030, Brazil

* Correspondence: fab.costa@usp.br

**Abstract:** Few studies have been published on the analysis and correlation of data from process mineralogical studies of gold ore employing artificial neural networks (ANNs). This study aimed to analyse and investigate the correlations obtained by the technological characterization of auriferous ore using an ANN called self-organizing map (SOM) to support geometallurgical studies. The SOM is a data analysis technique in which patterns and relationships within a database are internally derived and the outputs are visual, assisting in the understanding of data in the representation of 2D maps. In the representation generated, it was possible to establish that the variables of accessibility, exposed perimeter, median gold grain diameter (D50), and $SiO_2$ and arsenic contents have strong positive correlations. Regarding geometallurgy, this study shows that SOM can identify large-scale spatial chemical–mineralogical gold ore patterns, which can help define the most relevant indicator variables for mineral processing.

**Keywords:** gold process mineralogy; artificial neural network; geometallurgy





## 1. Introduction

In the mining industry, the processing of gold ores is directly related to their physical, chemical, and mineralogical characteristics which correspond to the performance of the ore in the beneficiation and extraction processes [1,2].

Process mineralogical studies are essential for recognising the intrinsic characteristics of the ore. They comprise the study of the properties of mineral raw materials, which are fundamental for the sustainable use of resources, providing information on the potential for material recovery, in addition to allowing the predictability of beneficiation processes and their waste management. Thus, the characterization of such materials focuses on obtaining parameters referring to the mineralogical assembly and its behaviour in the beneficiation process [3–7].

Based on the gold recovery and mineral processing techniques required, gold ores are commonly classified into two major groups: free-milling and refractory ores. Typically, free-milling ores are defined as those where over 90% of gold can be recovered by conventional cyanide leaching or some combination of flotation and cyanidation, while refractory ores are characterized by low gold recoveries using a significant number of reagents or a more complex pre-treatment process [8–11].

In this context, studies have focused on the characterization of gold grains through specific procedures that establish the main mineralogical parameters for processing, such as grain size distribution, characteristics of gangue minerals, mineral association, and accessibility [8,12–16].

Several advances in gold characterization have been made in the area of automated image analysis systems using scanning electron microscopes (SEM-AI), allowing the analysis of large groups of samples with speed, reliability, and robust results [17–19]. One of these systems, the Mineral Liberation Analyser (MLA), is commonly coupled to a scanning electron microscope (SEM) with energy-dispersive X-ray spectroscopy (EDS) to capture, store, and process raw data. It is a widely used technique for technological characterization in which it is possible to evaluate not only the punctual chemical composition of the mineral phases, but also the forms of mineralogical associations [20,21]. However, the concept of "liberation", one of the parameters analysed by the SEM-AI, loses its meaning [22] as the analysis process takes place through the exposed surface of the grain whose gold can be accessible through microfractures. The accessible portion is directly proportional to the ability to extract gold from a cyanide solution via fractures or microfractures. Figure 1 illustrates various occurrences of gold grains in relation to the exposure perimeter for cyanide solution percolation. While Figure 1A shows a free gold grain, Figure 1B concomitantly displays in the same single mineral a locked gold grain and gold that may be extracted by fracture for the solution percolation, making it accessible. Since it is a two-dimensional representation, it is evident that the gold may be extracted by paths located in another part of the grain not visualised in the 2D image. Thus, information on gold accessibility by image analysis may be undersized or underestimated due to pixel size resolution.

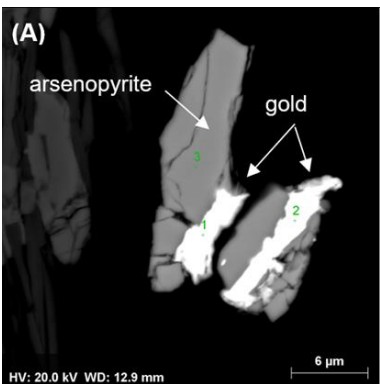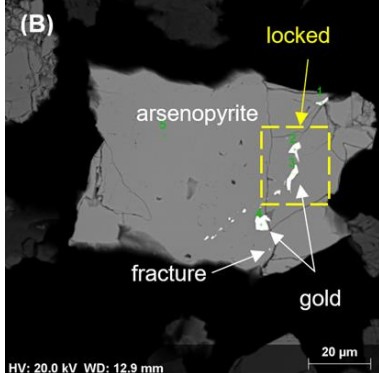

**Figure 1.** Accessibility of gold grain. (**A**) Gold grain exposure and (**B**) gold grain with minimum exposure and accessibility, liable to be leached, and gold grain locked in an arsenopyrite particle.

Due to the large volume of information generated by the mineralogical characterization, the self-organizing map (SOM) technique was applied. First proposed by Kohonen [23–25], this technique allows the visualisation and analysis of data based on the principles of vector quantisation and similarity measures. It can generate maps in an unsupervised and competitive way, preserving the topology between the input and output data of the networks. The main advantage of using an SOM system is its capability of handling several types of classification problems while providing a useful, interactive, and intelligible summary of the data. The reduced dimensionality and grid clustering facilitate the observation of similarities in the data. The SOM technique requires enough data to develop meaningful clusters, and the lack of data or extraneous data in the weight vectors tend to add randomness to the groupings, causing clustering to be incoherent.

Some disadvantages of the SOM approach are related both to the input variables and to the choice of the number of clusters. Redundant or uncorrelated variables can generate discrepancies that can hinder the observation of similarities in the U-matrix. Therefore, the use of analysis techniques to evaluate correlations between variables, e.g., PCA and grouping techniques, such as K-means clustering associated with the Davies–Bouldin index, can be helpful in SOM analyses.

The aim of this study is to analyse the correlations of the variables obtained by the technological characterization of auriferous ore, especially the variable of accessibility, using the SOM technique in the formation of clusters. This approach consists of grouping

the neurons that make up the self-organized map using the K-means algorithm [26,27], with the number of clusters defined based on the Davies–Bouldin index [28].

*Definition of Self-Organizing Maps*

The SOM technique consists of a multivariate analysis in artificial neural networks (ANNs) with unsupervised learning [29], that is, without human intervention during the model learning process and with little information about the characteristics of the input data. Unlike other ANN techniques, the SOM groups and reduces the dimensionality of the data, being capable of converting complex, high-dimensional, and nonlinear statistical relationships into simple geometric relationships with low-dimensional visualisation. In addition, this technique promotes unsupervised clusters with good visualisation of the relationships between variables in different types of data.

The implementation process of a data analysis routine using SOM presented by Fraser and Dickson [30] is performed by following two steps of data formatting and insertion into training bases. In the first step, training is carried out through a competitive learning process, also called "rough training", where an initial neighbourhood radius number, that is, the neurons neighbouring the best-adjusted neuron (i.e., the best-matching unit—BMU), is used, modifying in a single step a proportionally high number of neurons that make up the network. Then, a more refined step based on cooperative learning or "fine training" starts. It uses a smaller radius for the neighbourhood, modifying fewer neurons per interaction. Network training is a continuous process of comparing the prototype vectors of each neuron and the sample vectors that make up the database. Two SOM validation metrics are performed. The first is the quantisation error (qe), which is a measure of map resolution that also indicates how far apart the nodes are, while the second refers to the topological error (te), which translates the topological preservation of the data in the 2D map.

An important step is the choice of parameters related to training in an SOM environment. It initially involves choosing the size of the resulting self-organized map depending on the number of samples (N), with an area equal to $5 \times \sqrt{N}$ [31].

After all iterations are concluded, the best number of clusters of neurons is obtained based on the Davies–Bouldin index [28]. By combining the characteristics of the similarity measures, it indicates the similarities between the domains, inferring the adequacy of different data partitions regardless of the clustering technique used and the number of domains formed. Thus, the lower the Davies–Bouldin index, the more similar the domains are.

Another technique that is applied is the principal component analysis (PCA). It is most commonly used technique to reduce large datasets, facilitating the investigation of relationships among chemical, geological, and process variables. PCA is a linear technique whose visual results are essentially scatterplots that reflect linear correlations. In addition, it is a system that represents the data in n-dimensions by lines or planes (to whichever they have a better fit) whose main aim is to reduce data dimensionality with a large number of inter-variables related, retaining the variation present in the dataset as much as possible [32].

K-means clustering is also a widely used technique to identify groupings (clusters) in the SOM output neurons due to the efficiency of its algorithm and the simplicity of its application [33,34]. The algorithm is performed in two phases: the definition of the initial K centroids for each domain and the association of each point of the dataset with the closest centroid measured through the Euclidean distance. When all points are included in a group, the first phase is completed, and initial domains are formed. The centroids are recalculated, and new domains are formed with the new centroids until there are no more changes [34].

The application of analysis techniques and the input of variables in the SOM environment allowed the investigation of the broadest spectrum of possible correlations and therefore the identification of subtle characteristics and features in the joint analysis of the variables that make up the database, which would hardly be perceived in common data representations.

## 2. Materials and Methods

### 2.1. Input Dataset

The application of the SOM technique for the chemical–mineralogical characterization of gold ores obtained by MLA was carried out on the SiroSOM® application [35]. The samples were characterized at the Technological Characterization Laboratory (LCT) of the University of São Paulo (USP), Brazil. Figure 2 shows the flowchart of activities and procedures performed according to the methodology used.

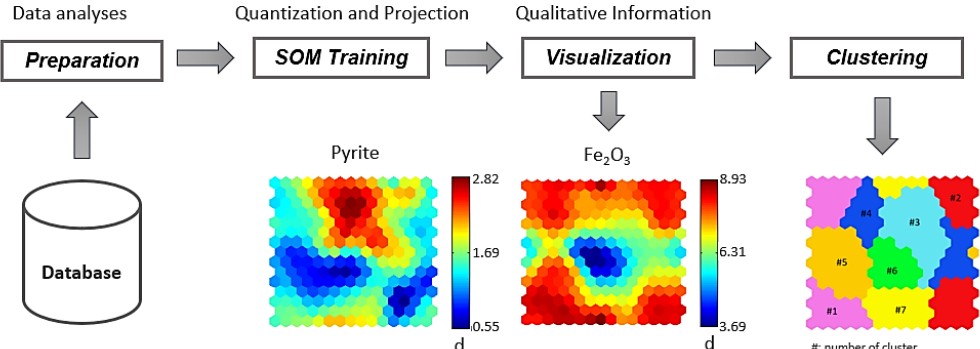

**Figure 2.** Flowchart of activities performed until clustering.

The processing starts with the input dataset of variables and their attributes. In this study, 22 variables of chemical–mineralogical characterization and their attributes were defined in the pre-processing phase.

The SOM training input data and the processing parameters can be seen in Table 1. The steps defined as "rough" and "fine training" are the trainings carried out through an interactive process comprising competitive and cooperative learning, respectively. These steps were applied several times for each sample unit until the best seed-vector represented the input data. The 2D map of the output data has the key feature of preserving the relative topological relationships between the node vectors.

**Table 1.** Parameters used for SOM training.

| Size Map | Rough Training | | | Fine Training | | |
|---|---|---|---|---|---|---|
| | IR * | FR * | FL * | IR * | FR * | FL * |
| 15 × 15 | 22 | 6 | 40 | 6 | 1 | 800 |

IR *: initial radius; FR *: final radius; FL *: final length.

The grid size equals the best approximation to a value five times the square root of the number of cases [32]. In this exploratory study, a map size of 15 rows × 15 columns was chosen as appropriate. When all iterations are concluded, the output is an organized, low-dimensional representation of the dataset, a bidimensional image represented by colour. The processing also creates a U-matrix as an output that represents the distance between neighbouring nodes for all attributes, with cold colours reflecting greater dissimilarities [36].

### 2.2. Variables and Reference Values

For the SOM analysis, the selected variables represent those most relevant to support geometallurgy. The variables that made up the database are listed in Table 2. They comprise the chemical analyses (oxides: X-ray fluorescence; Au: fire assay; S: induction furnace pyrolysis; and ICP-OES for As); the gold, arsenic, and sulphur contents; the mineralogical composition of the initial sample; the median equivalent diameter ($d_{50}$) for gold; the mineralogical association and its distribution; and the percentage of heavy products.

**Table 2.** Characterization variables used for SOM training.

| Variable | | Description | Variable | | Description |
|---|---|---|---|---|---|
| $Na_2O$ | % | sodium oxide | mica | % | min. distrib. mica |
| MgO | % | magnesium oxide | chlorite | % | min. distrib. chlorite |
| $Al_2O_3$ | % | aluminium oxide | albite | % | min. distrib. albite |
| $SiO_2$ | % | silicon dioxide | carbonates | % | min. distrib. carbonates |
| $K_2O$ | % | potassium oxide | pyrite | % | min. distrib. pyrite |
| CaO | % | calcium oxide | arsenopyrite | % | min. distrib. arsenopyrite |
| $Fe_2O_3$ | % | iron oxide | $d_{50}$ | μm | equivalent diameter $D_{50}$ in 2D |
| grade_As | ppm | arsenic grade | exposed perimeter | % | exposed perimeter association |
| grade_S | % | sulphur grade | accessible | % | accessible gold grain |
| grade_Au | g/t | gold grade | not encapsulated | % | not encapsulated gold grain |
| quartz | % | * min. distrib. quartz | locked | % | locked gold grain |

\* min. distrib: mineralogical distribution.

The reference values obtained from the original total dataset are presented in Table 3. These values were used to classify the contribution of each cluster derived from the components. The classification of each variable was performed based on the total values, which were ordered and divided into three quantiles (i.e., low, medium, and high), resulting in the generation of the reference values. Seven variables intrinsically related to the greater potential of gold performance in mineral processing were selected. The variable $Al_2O_3$ was highlighted due to the presence of gold grains being associated with aluminium silicate minerals such as chlorite and mica.

**Table 3.** Reference values to determine the classification of seven variables.

| | $Al_2O_3$ (%) | Accessible (%) | * Not Encap. (%) | Locked (%) | Exposed Perimeter (%) | Au Grade (g/t) | As Grade (ppm) |
|---|---|---|---|---|---|---|---|
| Minimum | 7.0 | 0.40 | 0.06 | 0.1 | 1.18 | 0.04 | 229 |
| Maximum | 22.4 | 100.0 | 74.2 | 100.0 | 100 | 2.12 | 6684 |
| Medium ($\bar{x}$) | 15.7 | 58.6 | 15.8 | 30.7 | 24.6 | 0.615 | 1989 |
| Median ($\tilde{x}$) | 15.6 | 64.0 | 9.50 | 23.5 | 16.3 | 0.500 | 1842 |
| SD (σ) | 3.35 | 30.7 | 18.0 | 26.4 | 22.4 | 0.440 | 1210 |
| Reference | | | | | | | |
| Low values | <14.0 | <41.8 | <7.0 | <15.1 | <10.7 | <0.334 | <1205 |
| Medium values | 14.0–17.1 | 41.8–79.9 | 7.0–41.5 | 15.1–36.9 | 10.7–24.8 | 0.334–0.638 | 1205–2268 |
| High values | >17.1 | >79.9 | >41.5 | >36.9 | >24.8 | >0.638 | >2268 |

\* Not encap: not encapsulated.

## 3. Results and Discussion

After testing a series of training steps for the SOM analysis using a 15 × 15 grid, the first vector training (rough training) was performed with 40 interactions followed by the second training (fine training) with 800 interactions. An average quantisation error of 2.58 and a topological error (te) of 0.037 were achieved in the formation of the SOM. The high variability, especially between the percentages of accessible and included gold, may explain the moderately high quantisation error.

The results obtained by the component planes that allow visualising and quantifying the contribution of the 22 input variables are shown in Figure 3. While Figure 4A provides a representation of a self-organizing map in the form of a U-matrix or unified distance matrix in terms of Euclidean distance, Figure 4B presents the classification of BMUs in seven domains based on K-means clustering.

The individual contributions of each variable highlight the relationships between the various components in a 2D-space visualisation. Thus, the 22 components (variables) are presented in colour temperatures ranging from the lowest (blue) to the highest value (red).

The U-matrix, also in colour temperature, represents the similarities between adjacent nodes in blue-green and the dissimilarities in orange-red.

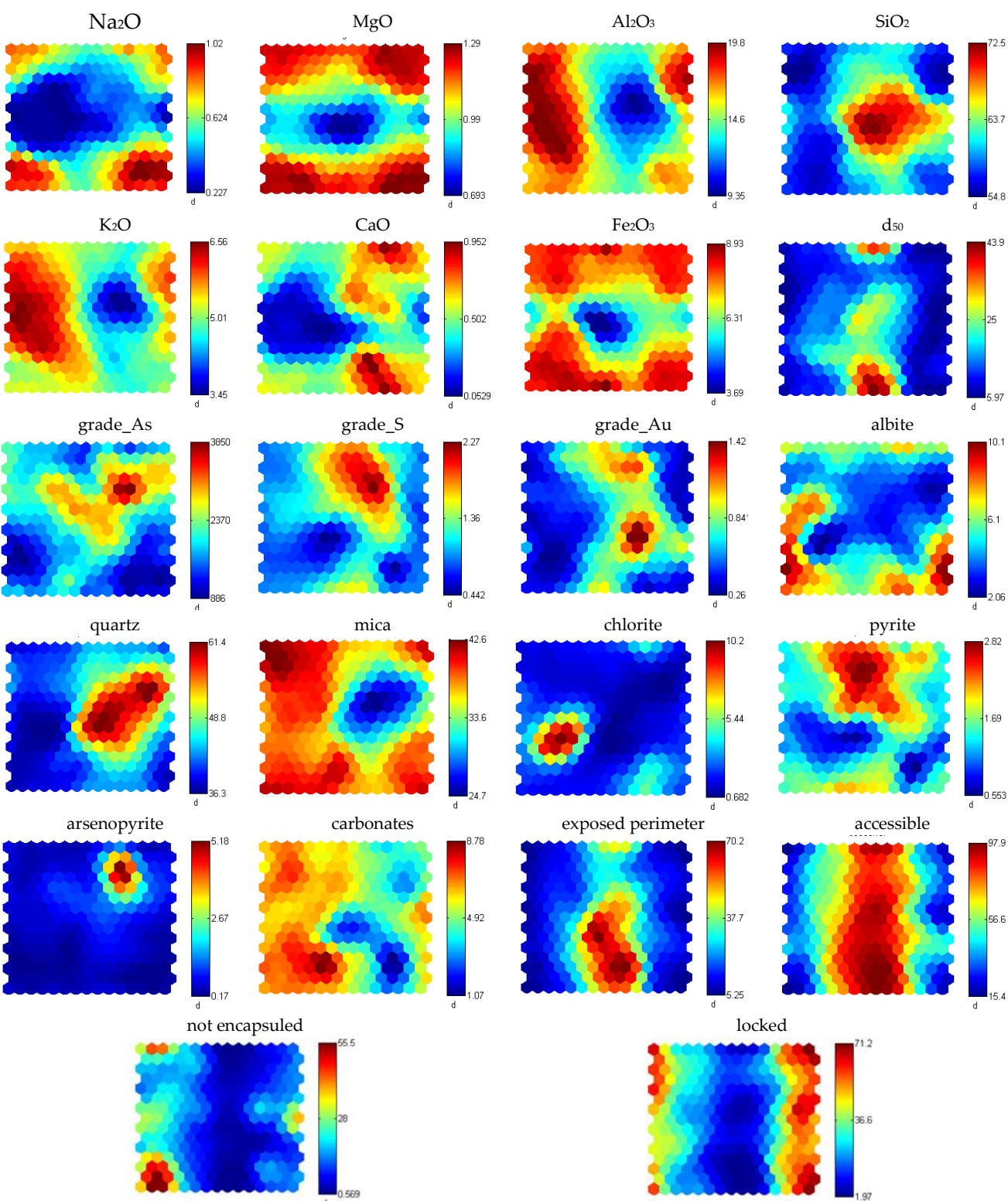

**Figure 3.** Representation of self-organizing maps and contributions of characterization variables.

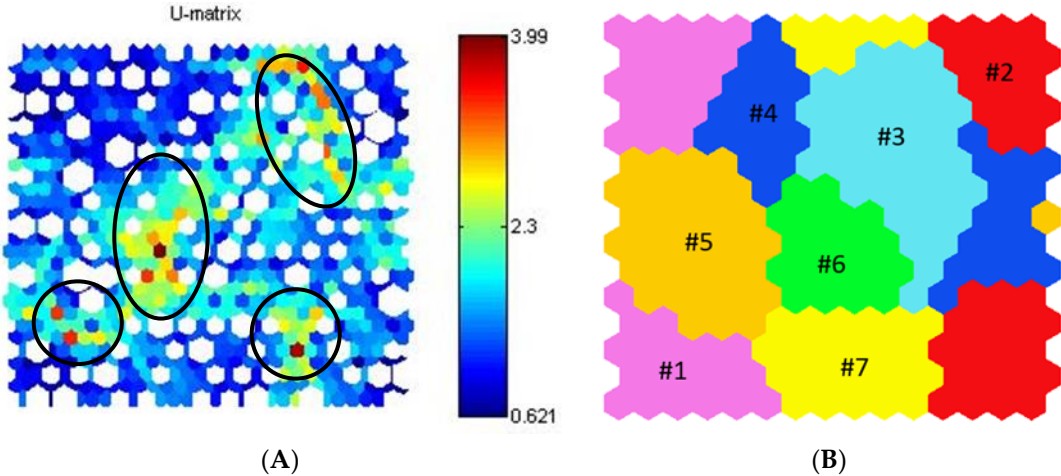

**Figure 4.** Representation of an SOM. (**A**) U-matrix and (**B**) classification of BMUs from K-means clustering. Note: (**A**) In the U-matrix, coloured neurons are used to represent the similarities (cold colours), whereas hot-coloured hexagons mean strong dissimilarity. The white hexagons are scaled proportionally to the number of samples; (**B**) Classification of BMUs in the self-organizing map from K-means. (# is number of cluster).

The U-matrix enabled the classification of data according to vector similarities constructed from these samples. It shows four main areas with high dissimilarity, highlighted in Figure 4A by circles, which coincide with elevated contributions of $SiO_2$, grade_S, quartz, and chlorite. The high similarity in the U-matrix is associated with strong contributions of grade_Au, $d_{50}$, arsenopyrite, accessible gold, and exposed perimeter. The topological distribution of the values in the SOM map reveals a trend of distribution of the highest values along a central band.

In the representation of self-organizing maps, the contributions of multiple variables have strong positive correlations between the variables. Based on the observation of these maps, the variable of accessibility, which corresponds to the gold grains containing a gold parcel that can be extracted, shows a good similarity with $d_{50}$, a variable that refers to the equivalent circle diameter; exposed perimeter; and $SiO_2$ and arsenic contents. The relationship with exposed perimeter of the grains confirms the relationship between greater potential for gold extraction and higher percentage of exposed perimeter as well as larger gold diameter. The term exposed perimeter is defined as the perimeter of a gold grain that is in contact with no other grain. The accessibility of gold is directly related to the gold's possibility of being extracted by a leaching solution, for example [37].

Oxides, mainly $Na_2O$, $MgO$, $K_2O$, and $Fe_2O_3$, did not show any similarity with potential variables that indicate gold recovery, such as accessibility and exposed perimeter. The results also revealed a relationship between gold grains associated with arsenopyrite and pyrite.

PCA was applied to analyse the distribution of scores (PC1 and PC2) in the 2D space. According to the results, within the domains formed, there are four sets of well-defined variables (Figure 5), two of which (the quadrants indicated as III and IV) correlate with a higher gold extraction potential and higher Au, As, and S contents. The occurrence of gold, as shown in these sets, is mostly associated with arsenopyrite and pyrite and smaller amounts of sphalerite, galena, and chalcopyrite, represented by the variable, other sulphides. Quadrants I and II indicate similarities between silicate minerals and their relationship with encapsulated and included gold grains with low recovery potential and the association of gold grains with silicates such as chlorite, mica, and albite and carbonates represented by ankerite and dolomite.

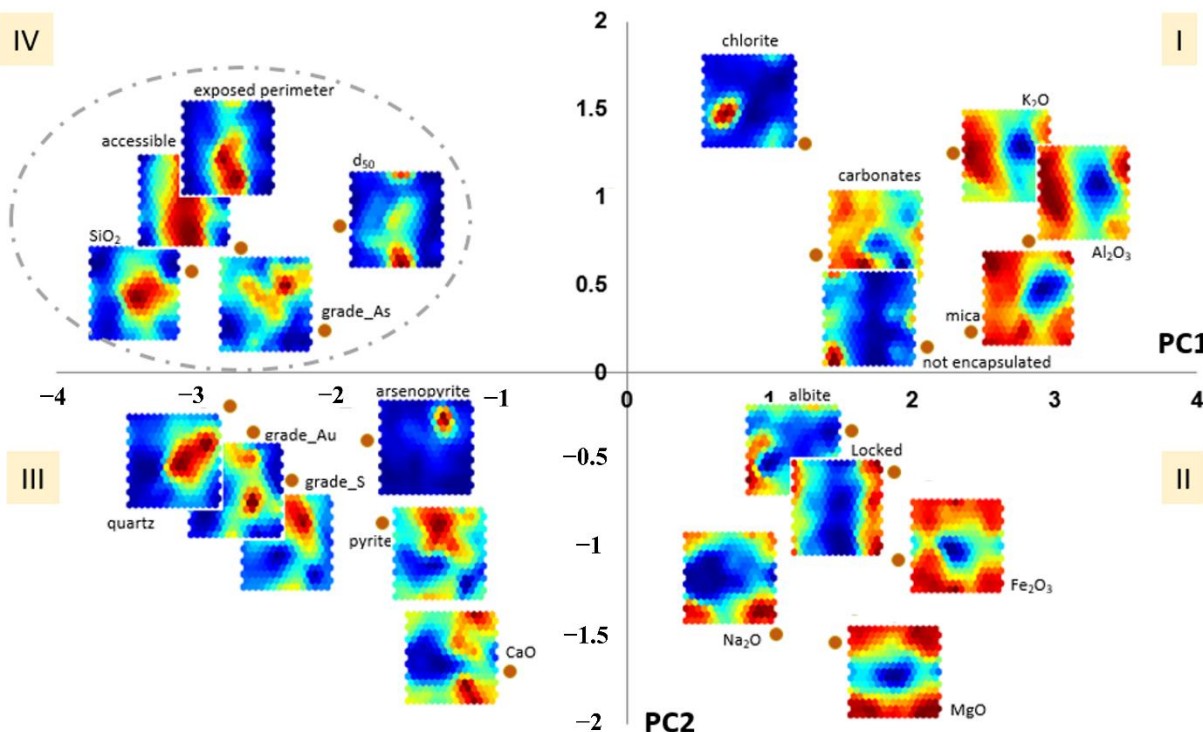

**Figure 5.** Projection of the main components: representation of the SOM and distribution of scores. Note: Representation of the contributions of chemical and geological variables. Distribution of scores of the 22 variables in the principal components (PC1 and PC2), which are created in order of variation: PC1 captures the greatest variation, while PC2 detects the second greatest variation. I, II, III and IV means the quadrants and their groupings.

Davies–Bouldin analysis was performed to define the ideal number of clusters for neurons from the SOM analysis. For the representation of the BMU clusters, seven groups were obtained within a classification based on K-means, which shows the spatial distribution of the input samples assigned to each resulting SOM cluster. Table 4 summarises the influence of the most relevant variables on the seven clusters thus defined.

**Table 4.** Influence of variables on each cluster of SOM analyses.

| | | $Al_2O_3$ (%) | Accessible (%) | Not Encap. (%) | Locked (%) | Exposed Perim. (%) | Grade_Au (g/t) | Grade_As (ppm) |
|---|---|---|---|---|---|---|---|---|
| Cluster 1 N: 41 | * BMU Influence | 17.0 Medium | 44.5 Medium | 15.3 Medium | 43.9 High | 13.1 Medium | 0.528 Medium | 2160 Medium |
| Cluster 2 N: 33 | * BMU Influence | 16.6 Medium | 37.0 Low | 24.4 Medium | 43.5 High | 11.0 Medium | 0.465 Medium | 1340 Medium |
| Cluster 3 N: 18 | * BMU Influence | 13.2 Low | 74.1 Medium | 6.1 Low | 22.0 Medium | 25.2 High | 0.950 High | 2200 Medium |
| Cluster 4 N: 21 | * BMU Influence | 14.4 Medium | 91.1 High | 3.6 Low | 7.3 Low | 52.3 High | 0.765 High | 1670 Medium |
| Cluster 5 N: 9 | * BMU Influence | 13.1 Low | 90.2 High | 4.5 Low | 8.1 Low | 56.0 High | 0.770 High | 2520 High |
| Cluster 6 N: 14 | * BMU Influence | 10.6 Low | 76.4 Medium | 8.2 Medium | 17.2 Medium | 32.6 High | 0.916 High | 2850 High |
| Cluster 7 N: 27 | * BMU Influence | 18.8 High | 52.4 Medium | 20 Medium | 29.3 Medium | 19.3 Medium | 0.381 Medium | 1740 Medium |

* BMU: best-matching unit; Not encap.: not encapsulated. Exposed perim.: exposed perimeter.

As observed, Cluster 1 has a high gold content that is possibly included in arsenopyrite and pyrite since it shows a high arsenic content. The 18 samples in this cluster represent high values of non-encapsulated and included gold grains, implying that low recovery is expected in the mineral processing. Although Cluster 2 has the same characteristics as Cluster 1, it has a lower arsenic content along with low accessibility.

Clusters 3 and 6 show similar behaviours, with negligible variations in the high levels of arsenic. What differs is the $Al_2O_3$ content, which possibly presents variations in the content of silicate minerals. Cluster 4 is defined by high accessibility, high gold content, and medium arsenic content. This type of cluster is characteristic of samples from more weathered regions where a low content of $Al_2O_3$ is found. Cluster 7, where most samples occurred, is composed of samples with high $Al_2O_3$ content, high accessibility, and average gold and arsenic contents. This cluster may represent the deposit in terms of average indices and average gold content, which is 0.379 ppm.

## 4. Conclusions

By using the integrated dataset obtained by the chemical–mineralogical characterization of gold ores, it was possible to establish in the representation of self-organizing maps that the contributions of the multiple variables had strong positive correlations, mainly accessibility, $d_{50}$, exposed perimeter, and $SiO_2$ and arsenic contents.

The results obtained from the SOM analysis indicated the formation of seven clusters, which were related to different grades, and defined the influence of the most relevant variables.

Oxides such as $Na_2O$, $MgO$, $K_2O$, and $Fe_2O_3$ did not show any similarities with potential variables that indicate gold recovery, such as the variable of accessibility. The application of PCA revealed the presence of four sets of well-defined variables, two of which (I and II) correlated with a greater potential for gold extraction and Au, As, and S contents. Quadrants III and IV showed similarities between silicate minerals and encapsulated and included gold grains with low recovery potential as well as an association of gold grains with silicates and aluminium-silicates.

In the Davies–Bouldin analysis based on K-means, seven clusters were defined according to their similarities. The results pointed to the existence of a tenuous threshold between the grades of each cluster that is difficult to observe due to the little lithological variation.

From the point of view of geometallurgy, the definition of the most relevant indicator variables for mineral processing, in addition to the variables of greater similarity, demonstrates a significant advance in data integration as it opens the possibility of better interpreting and understanding their performance in the construction of a geometallurgical predictive model.

**Author Contributions:** F.R.C.—methodology, investigation, writing—original draft. C.d.C.C.—conceptualization, writing—review and editing, validation. C.U.—conceptualization, methodology, resources, writing—review and editing, project administration, funding, writing—review and editing. All authors have read and agreed to the published version of the manuscript.

**Funding:** The infrastructure was provided by LCT and InTRA Laboratories. The scholarship provided to F.R.C. M.T. was granted by the Coordination for the Improvement of Higher Education Personnel (CAPES), financé code 001.

**Data Availability Statement:** Data supporting the findings of this study will be made available from the corresponding author upon reasonable request.

**Acknowledgments:** We would like to thank the technical team of the Technological Characterization Laboratory (LCT) at the Polytechnic School of USP for their analytical support and InTRA for the software availability. We are grateful to FINEP (grants 01.18.0041.00 and 01.18.0137.00) and FAPESP (grants 2020/06754-0 and 2020/08476-8) for the infrastructure provided and CAPES for the scholarship offered to F.R. Costa. The authors also wish to thank the anonymous referee for reviewing the manuscript and providing valuable comments and suggestions.

**Conflicts of Interest:** The authors declare no conflict of interest.

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
