# Peer review of "Self-Organizing Maps Analysis of Chemical–Mineralogical Gold Ore Characterization in Support of Geometallurgy"

_mining, doi:10.3390/mining3020014_

Round 1

Author Response

São Paulo, March 30th  2023

Review report for manuscript Mining-2254678

Self-organizing map analysis of chemical-mineralogical gold ore characterization in support to geometallurgy

To

Dr. Chuthathip Mangkonsu

Assistant Editor

Dear Dr. Mangkonsu

I kindly ask you to see below the authors' considerations about the points observed by the reviewers. I would like to thank the reviewers for their contribution to the manuscript. Several changes and additions have been made.

  1. On page 1, lines 21-22, I believe some words are missing.

Authors: Revised

  1. Typos on page 1, line 29 (essential); Page 4, line 150 (FL*).

Autors: Revised

  1. On page 1, line 46, SEM was not defined during the initial appearance.

Authors: Revised

  1. There are multiple mismatches of figure numbers. I think Fig. 2, 3, and 4 mentioned in

the paragraphs are Fig.3, 4, and 5.

Authors: Revised

  1. Citation 36 is not accessible.

https://www.csiro.au/en/work-with-us/industries/mining-resources/exploration/self-organising-maps

Authors: Accessed according to citation

  1. On page 7, lines 203-204, I don’t think there is a strong relationship between “accessible”

and “d50”, at least based on the images the manuscript shows. Instead, I believe

“accessible” has a strong co-relationship with “exposed perimeter.” Can the authors give a further explanation?

Authors: The topological distribution of values in the SOM map shows a trend of distribution of the highest values along a central band.

There is a strong relationship between d50, exposed perimeter and accessible.

  1. I recommend further editing Fig. 5. Quadrant I should be at the top right corner, and

Quadrant II should be at the bottom right corner. So on and so forth

Authors: Revised

Best regards,

Fabrizzio Costa, Cleyton Carneiro, Carina Ulsen

Reviewer 2 Report

Authors propose the construction of a geometallurgical predictive model by the correlations obtained by technological characterization of auriferous ore.

The manuscript reports elaborations of experimental results aimed at evaluating the best predictive model. The submitted manuscript is of sure interest in geometallurgy.

However, the description is not always easy to follow and I believe that the authors should better organize the results and discussion to help less experienced readers understand how the most appropriate predictive models can be evaluated.

I suggest authors:

- English language should be improved

- Results and discussion should be better organized and the correlations obtained should be made more accessible to non-expert readers. In this view, authors should have colleagues read the manuscript to evaluate the degree of reading difficulty of the discussion

Author Response

São Paulo, March 30th  2023

Review report for manuscript Mining-2254678

Self-organizing map analysis of chemical-mineralogical gold ore characterization in support to geometallurgy

To

Dr. Chuthathip Mangkonsu

Assistant Editor

Dear Dr. Mangkonsu

I kindly ask you to see below the authors' considerations about the points observed by the reviewers. I would like to thank the reviewers for their contribution to the manuscript. Several changes and additions have been made.

Comments and Suggestions for Authors

Authors propose the construction of a geometallurgical predictive model by the correlations obtained by technological characterization of auriferous ore.

The manuscript reports elaborations of experimental results aimed at evaluating the best predictive model. The submitted manuscript is of sure interest in geometallurgy.

However, the description is not always easy to follow and I believe that the authors should better organize the results and discussion to help less experienced readers understand how the most appropriate predictive models can be evaluated.

I suggest authors:

- English language should be improved.

English review will be performed.

- Results and discussion should be better organized and the correlations obtained should be made more accessible to non-expert readers. In this view, authors should have colleagues read the manuscript to evaluate the degree of reading difficulty of the discussion

A detailed review was carried out in order to make the text more comprehensive for non-expert readers. Some definitions have been introduced to make the manuscript clearer.

Best regards,

Fabrizzio Costa, Cleyton Carneiro, Carina Ulsen

Reviewer 3 Report

This paper applies the self-organizing map (SOM) method to the unsupervised classification of gold ore samples based on 22 parameters.

Even if the advantages of the SOM method are not so obvious in the case of the gold ore considered in this article, the latter nevertheless has the merit of contributing to popularize this method in the field of geosciences.  The results are coherent and the authors seem to have a good command of the numerical techniques used. I therefore recommend the publication of this article in Mining.

However, in its current form, this article appears to have a few shortcomings that will need to be addressed if the editor approves the publication of this article. In particular:

The English should be thoroughly revised, as several sentences are incomplete or poorly constructed, making the text difficult to understand in places.

Figures are not properly cited in the text.

The captions for Figures 4 and 5 are unusual. They should begin with Figure X ...

The advantages (and disadvantages) of the SOM method over other popular unsupervised classification methods, such as PCA or K-mean (speed, representativeness, etc.), should be discussed in the introduction.

At the beginning of the article, information should be provided about the mining samples used, especially their provenance, number, and the reason why they were selected for this study.

It is mentioned that the dimension of the self-organizing map is 5√N = 15x15, where N is the number of samples. However, it does not appear that this expression is consistent with the number of samples that is 163 by summing the group members in Table 4.

The U-matrix in Figure 4(a) should be explained in more detail. Shouldn't SOM neurons be identified in this figure? The meaning of the white hexagons is not clear to me. The identification of the circles mentioned in the caption is not clear either. Are they the red and orange dots?

There seems to be some similarity between the areas in Figure 4(a) and those in Figure 4(b). Is this expected or is it a coincidence?

It would have been interesting to examine a smaller number of K-mean domains, as it seems that 7 is excessive. From Table 4, the differences between some groups do not appear to be significant.

Author Response

São Paulo, March 30th  2023

Review report for manuscript Mining-2254678

Self-organizing map analysis of chemical-mineralogical gold ore characterization in support to geometallurgy

To

Dr. Chuthathip Mangkonsu

Assistant Editor

Dear Dr. Mangkonsu

I kindly ask you to see below the authors' considerations about the points observed by the reviewers. I would like to thank the reviewers for their contribution to the manuscript. Several changes and additions have been made.

Comments and Suggestions for Authors

This paper applies the self-organizing map (SOM) method to the unsupervised classification of gold ore samples based on 22 parameters.

Even if the advantages of the SOM method are not so obvious in the case of the gold ore considered in this article, the latter nevertheless has the merit of contributing to popularize this method in the field of geosciences.  The results are coherent and the authors seem to have a good command of the numerical techniques used. I therefore recommend the publication of this article in Mining.

However, in its current form, this article appears to have a few shortcomings that will need to be addressed if the editor approves the publication of this article. In particular:

The English should be thoroughly revised, as several sentences are incomplete or poorly constructed, making the text difficult to understand in places.

Figures are not properly cited in the text.

Authors: Revised

The captions for Figures 4 and 5 are unusual. They should begin with Figure X ...

Authors: Revised

The advantages (and disadvantages) of the SOM method over other popular unsupervised classification methods, such as PCA or K-mean (speed, representativeness, etc.), should be discussed in the introduction.

Authors: Added lines 71-81.

At the beginning of the article, information should be provided about the mining samples used, especially their provenance, number, and the reason why they were selected for this study.

Authors: Unfortunately, due to confidentiality reasons, we cannot provide information on where the samples come from.

It is mentioned that the dimension of the self-organizing map is 5√N = 15x15, where N is the number of samples. However, it does not appear that this expression is consistent with the number of samples that is 163 by summing the group members in Table 4.

Authors: Adjustments were made to have a good dimensionality of the maps.

The U-matrix in Figure 4(a) should be explained in more detail. Shouldn't SOM neurons be identified in this figure? The meaning of the white hexagons is not clear to me. The identification of the circles mentioned in the caption is not clear either. Are they the red and orange dots?

Authors: Adjusted the caption to be clearer.

“Note: (A) U-Matrix, colored neurons to represent their respective similarities (cold colors); Hot colors hexagons means strong dissimilarity. The white hexagons represents scaled proportionally to the number of samples; (B) Classification of BMUs on the self-organizing map from the K-means.”

There seems to be some similarity between the areas in Figure 4(a) and those in Figure 4(b). Is this expected or is it a coincidence?

Authors: The figure 4A represents the U-Matrix, where colored hexagons represents similarities and dissimilarities. The Figure 4B, show the classification of BMUs on the self-organizing map from the K-means. The figures must be similar.

 It would have been interesting to examine a smaller number of K-mean domains, as it seems that 7 is excessive. From Table 4, the differences between some groups do not appear to be significant.

Authors: Future tests will be carried out to evaluate the representativeness of the clusters.

Best regards,

Fabrizzio Costa, Cleyton Carneiro, Carina Ulsen

Round 2

Reviewer 2 Report

Authors modified as suggested.

Submission can be accepted